# Diamondback Moth Larvae Trigger Host Plant Volatiles that Lure Its Adult Females for Oviposition

**DOI:** 10.3390/insects11110725

**Published:** 2020-10-23

**Authors:** Mubasher Hussain, Jing Gao, Summyya Bano, Liande Wang, Yongwen Lin, Steve Arthurs, Muhammad Qasim, Runqian Mao

**Affiliations:** 1Guangdong Key Laboratory of Animal Conservation and Resource Utilization, Guangdong Public Laboratory of Wild Animal Conservation and Utilization, Guangdong Engineering Research Center for Mineral oil pesticides, Institute of Zoology, Guangdong Academy of Science, Guangzhou 510260, China; summyyasami@gmail.com; 2State Key Laboratory of Ecological Pest Control for Fujian and Taiwan Crops, Key Laboratory of Biopesticide and Chemical Biology, Ministry of Education (MoE), Fujian Agriculture and Forestry University, Fuzhou 350002, China; 3Zhangzhou Institute of Technology, Zhangzhou 363000, China; 18605062536@126.com; 4BioBee, Tucker, GA 30084, USA; stevenarthurs55@gmail.com; 5Centre of Agricultural Biochemistry and Biotechnology (CABB), University of Agriculture Faisalabad (UAF), Faisalabad 38000, Punjab, Pakistan; mqb4uaf@gmail.com; 6School of Life Sciences, Sun Yat-Sen University, Guangzhou 510275, China

**Keywords:** crucifer crops, diamondback moth, insect-plant interaction, HIPVs, volatile organic compounds, oviposition

## Abstract

**Simple Summary:**

The diamondback moth is a serious pest of crucifer crops. To control this pest, the use of intercropping (such as trap crops) is gaining attention since they are ecologically safe. However, such approaches require an understanding host plants which are most attractive to the pest. In this study, we quantified different volatiles released by *Barbarea vulgaris* in response to the diamondback moth larval feeding. We investigated olfactory responses of the adult diamondback moth to natural and simulated volatiles released by infested plants. We also investigated how volatile effects changed in response to larval feeding treatments. Overall, our findings indicated the relationship between key volatile compound, host plant cues emission and regulation of the diamondback moth adult female behavior due to key volatile triggered by the diamondback moth larvae feeding on *B. vulgaris*.

**Abstract:**

The diamondback moth (DBM) is a destructive pest of crucifer crops. In this study, DBM larvae shown to herbivore induced plant volatiles (HIPVs) that were attractive to adult females exposed in a Y-tube olfactometer. Our results showed that olfactory responses of adult females to HIPVs induced by third instar larvae feeding on *Barbarea vulgaris* were significantly higher (20.40 ± 1.78; mean moths (%) ± SD) than those induced by first instar larvae (14.80 ± 1.86; mean moths (%) ± SD). Meanwhile, a significant concentration of Sulphur-containing isothiocyanate, 3-methylsulfinylpropyl isothiocyanate, and 4-methylsulfinyl-3-butenyl isothiocyanate were detected in HIPVs released by third instar larvae compared to those released by first instar larvae while feeding on *B. vulgaris*. When the DBM females were exposed to synthetic chemicals, singly and in blend form, a similar response was observed as to natural HIPVs. Our study demonstrated that the relationship between isothiocyanates acting as plant defense compounds, host plant cues emission and regulation of the DBM adult female behavior due to key volatile triggered by the DBM larvae feeding on *B. vulgaris*.

## 1. Introduction

Plants are continuously at risks of attack by herbivore insect. As a result, they have evolved many inducible defense mechanisms to avoid damage [1]. For example, herbivory may modify the reallocation of primary plant metabolites or trigger other resistance-related plant responses in undamaged neighbors which act directly or indirectly against herbivores [2,3]. Direct defenses include a diverse array of strategies, such as strengthening of plant cell walls by regulating secondary metabolites, induction of hypersensitive cell death, and production of toxic and deterrent substances, such as glucosinolates and saponins in crucifers [4,5,6,7]. Indirect defenses can involve the emission of volatile compounds in response to arthropod feeding, which can influence trophic guilds by attracting natural enemies [8,9].

The diamondback moth (DBM), *Plutella xylostella* L., is a serious pest of crucifer crops with a cosmopolitan distribution [10]. The DBM has developed resistance to many chemical insecticides, as well as Bt toxins [11] making it difficult to control [12]. Efforts to find alternative control approaches include cultural methods, such as the use of trap crops and intercropping [11]. Such approaches require an understanding of competing host plants which are most attractive to the pest.

Host plant volatiles play an important role in host finding, recognition and acceptance by herbivores [13,14,15,16,17,18]. The brassica family produces a wide range of compounds which are protective against polyphagous species [19]. However, some of these secondary metabolites, such as glucosinolates, are used for host recognition cues by specialist Lepidoptera, including the DBM [19,20,21]. In addition, some plant volatiles are induced HIPVs, which can mediate ecological functions, including directly inhibiting development of herbivores and pathogens [11,14,15]. 

In this study, we hypothesize that host attraction by the DBM may reflect upregulated levels of key volatile compounds. Thus, we quantified different volatiles released by brassica plants in response to the DBM larval feeding. We investigated olfactory responses of adult the DBM to natural and simulated volatiles released by infested plants. We also studied how volatile effects changed in response to different larval feeding treatments.

## 2. Materials and Methods

### 2.1. Plants and Insects

Seed of wintercress (Barbarea vulgaris) (G-type R. Br, glabrous type; Hedeland population) obtained from the Department of Plant and Environmental Sciences, University of Copenhagen, Denmark were stored at 4 °C until use. Seeds were sown in a greenhouse under long-day conditions, 25 °C (light) and 20 °C (dark) with 16:8-h photoperiod and 60–75% relative humidity (RH). Plants were watered weekly and fertilized every other week. After two weeks, 120 seedlings were transplanted individually into 1-L plastic pots and maintained in climate-controlled room at 25 °C (light), 20 °C (dark), 16:8 h photoperiod and 60–65% RH.

*P. xylostella* L. (strain Fuzhou-S) larvae were collected from a brassica field in Fuzhou in southeastern China (26.08° N, 119.28° E), and reared on an artificial diet [22], in a climate-controlled room maintained at 24 ± 1 °C (light) and 24 ± 1 °C (dark) with a 12:12 h photoperiod and 60–70% RH. Emerging adult moths were released into mesh cages (60 × 60 × 60 cm) in groups of 400 (50:50 sex ratio) for mating. Adult female moths were used in the experiments

### 2.2. Behavioral Response Treatments

B. vulgaris plants (60–70 day old) were randomly assigned to three comparison groups with the DBM adult females for olfactory response, (1) healthy (HB) control versus mechanically damaged plants (MB), (2) plants exposed to first instar (FLB) versus third instars (TLB), and (3) first instars reared on artificial diet (FLA) versus first instar larvae (FL). Mechanical damage was induced by cutting across 30% of the B. vulgaris leaf veins with a sterilized knife. In each treatment, the DBM larvae were released on plants using a camel hair brush. There were four replicates for each treatment.

### 2.3. Olfactory Responses to Volatiles

Three treatment combinations were used to test the attraction of the chemicals to mated DBM adult female with an olfactometer: (1) healthy (HB) control versus mechanically damaged plants (MB), (2) plants exposed to first instar (FLB) versus third instar (TLB), and (3) first instars reared on artificial diet (FLA) versus first instar larvae (FL). The preference of the mated DBM adult female for HIPVs was studied in two-choice tests with a closed system Y-tube olfactometer, using the method described by Lin et al. [23]. The test chamber consisted of a Y-shaped glass tube (base tube: 10.5 cm; Y arms: 10 cm; tube internal diameter: 1.6 cm, and 90° angle between the two arms). Each arm was connected to a flow meter and an odor source container consisting of a glass jar (30 cm high, 28 cm diameter) large enough to hold a potted *B. vulgaris* plant. Parafilm was used to cover the soil of the pot, isolating it from the plant foliage. The airflow was purified through a charcoal filter and then passed through a humidifier bottle. The humidified airflow was split between two channels, and each channel was directed through an odor container. The two odor flows were sent through each arm of the olfactometer. The airflow through each arm was 300 mL/min as verified by a flow meter. The experiment was conducted at 20–25 °C with 60–70% RH, in a black box with an artificial light source from a single 35 W fluorescent tube placed above the arms of the Y-tube.

For olfactory responses, mated adult females were released individually at the open end of the Y-tube (joint arm). The time spent by adult females in each arm was recorded over 300 s, which began after an adult female touched the cotton placed at the end of the Y-tube arm. The olfactometer treatments were alternated between arms every five tests to prevent locational bias. Every 10 trials, the olfactometer tube was washed with alcohol and dried, and the two plants were replaced. Fifty adult females were tested per sets of plant treatments with a new unexposed adult female used for each trial replicate. In total, there were five replicates in this experiment.

### 2.4. Headspace Collection and Volatiles Analysis

We investigated the production of isothiocyanate (SIT) levels in response to larval feeding as potential host-attraction cues. The DBM larvae (3rd instar larvae) were infested on the surface of the plants with a camel hair brush. Volatiles were collected (after 24 h) from the olfactometer headspace as described by Pineda et al. [24]. The four treatments were: healthy *B. vulgaris*; *B. vulgaris* fed by first instar larvae (n = 50); *B. vulgaris* fed by third instar larvae (n = 50); and mechanically damaged *B. vulgaris* (n = 4).

For the assessment, each plant pot was wrapped with parafilm (Neenah, WI54956, USA) to prevent the soil odor from mixing with plant odor. Plants were placed in glass jars (10 L, cleaned with acetone, which was allowed to evaporate, and heated to 70 °C overnight before use). Pots containing soil alone were used as control treatments. Air was drawn from the glass jar through a glass tube (charcoal filter trap column, 3-mm ID, 65 mm long, 1.5 mg Prec charcoal, Klaüstrott, Chromatographie, Deerfield, MA, USA) using a mini vacuum pump (CD12/16NK, 1KL, Billerica, MA, USA), connected to the glass jar by a mini steel tube (2 mm ID) at 30 mL/min (negative pressure) for 24 h. Before use, the filter trap column was washed with dichloromethane-methanol (1:2 *v*/*v*), chloroform, acetone, dichloromethane, and n-pentene (in that order) and then heated to 100 °C for 12 h. All experiments were performed in a temperature-controlled room (24 ± 1 °C) with three replicates (n = 50) per treatment [23]. Collected volatiles were absorbed using Tenax^®^ TA 200 mg (60/80 mesh; Grace-Alltech, Deerfield, IL, USA), eluted into 30% chloroform, and the mixture was subsequently stored at −20 °C until analysis.

The adsorbed samples were analyzed using a gas chromatograph (Agilent Technologies (Santa Clara, CA, USA) 7890B GC System) mass spectrometer (Agilent Technologies 5977A MSD) (GC-MS) with an HP-5 column (30 mm × 0.25 mm i.d., 1.0 μm film thickness, Agilent). The GC oven temperature was increased from 40 °C (5-min hold) to 280 °C at a rate of 10 °C per min. Column effluent was ionized by electron impact ionization at 70 eV. Mass spectra were acquired by scanning from 35 to 350 m/z at a scan rate of 5.38 scans/sec.

Compounds were identified using the deconvolution software AMDIS (version 2.64, NIST, USA) in combination with NIST 05 and Wiley 7th edition spectral libraries, and by comparing retention indices with those from the literature.

### 2.5. Adult Female Response to Synthetic HIPV Sources

The response of the adult female DBM was compared between natural and synthetic HIPVs sources. For testing synthetics of active chemicals (sulfur-containing isothiocyanate, 3-methylsulfinylpropyl isothiocyanate, and 4-methylsulfinyl-3-butenyl isothiocyanate volatile compounds from the headspace) were purchased from Macklin Biochemical Co. Ltd. (Shanghai, China) at 97–99% purity.

For testing single doses, 3-methylsulfinylpropyl isothiocyanate were dissolved with triethyl citrate (TEC) to 0.01, 0.5, 5, 10, 20, and 30 nmol·mL^−1^. In addition, 4-methylsulfinyl-3-butenyl isothiocyanate was dissolved with TEC to 0.1, 0.5, 1, 5, 10, and 25 nmol·mL^−1^ concentrations. The range of dilutions used for the bioassay varied from Quantities of chemicals emitted from cotton roll to Quantities detected in *B. vulgaris* by GC-MS (Appendix A). For testing blends, 3-methylsulfinylpropyl isothiocyanate and 4-methylsulfinyl-3-butenyl isothiocyanate were tested at the following ratios; blend HB (18.71 and 0.10 nmol·mL^−1^, respectively), blend MB (8.32 and 1.36 nmol·mL^−1^, respectively), blend FLB (22.91 and 1.89 nmol·mL^−1^, respectively) and blend TLB (18.78 and 1.43 nmol·mL^−1^, respectively (Appendix A). Each suspension solution (1 mL) was applied to a small cotton roll and placed into the volatiles collecting chamber. The response of mated adult DBM to each blend were tested in the Y-tube olfactometer as described above. The selected adult females were starved for 24 h before the behavioral test, and the test was replicated five times.

### 2.6. Statistical Analysis

Responses of adult females to HIPVs in the five treatments were compared using chi-square tests for dependent samples with five replicates. We performed Pearson Correlation, the p-value for each test (not merely “significant” or “p < 0.05”), to determine the correlation between the olfactory responses of the adult DBM females to B. vulgaris volatile blends. We used SPSS 19.0 for overall statistical analyses.

## 3. Results

### 3.1. Quantities of Isothiocyanate Detected from the Headspace of Multiple Damaged Barbarea vulgaris Plants

The DBM larval feeding affected the levels of isothiocyanate volatiles produced by *B. vulgaris* plants. In the FLB and TLB treatments, means of 3-methylsulfinylpropyl isothiocyanate were significantly lower, while the MB treatments were not significantly different from the HB (control). (Table 1). These findings demonstrated that the FLB suppressed the emission of this sulphur containing isothiocyanates (SIT) volatile compound. The first instar feeding caused a decreased attraction of adult females to host plants (Table 2) (Mean quantity (ng) ± SE; 0.12 ± 0.03 ^c^). In contrast, third instar larvae feeding on *B. vulgaris* did not affect the attraction of adult females to host plants when compared with first instars (Figure 1).

### 3.2. Y-Tube Olfactometer Bioassays

Our Y-tube olfactometer bioassay responses indicated that HIPVs affected the behavior of adult female moths. Firstly, healthy plants were more attractive to the DBM adult female compared with mechanically damaged plants (Table 3). Moreover, plants infested with first instars were less attractive to the DBM adult females than plants infested with third instars or healthy plants (Table 3).

### 3.3. Adult Female Responses to Synthetic Compounds

The responses of the adult female DBM to single dosages and blends of 3-methylsulfinylpropyl isothiocyanate and 4-methylsulfinyl-3-butenyl as odor sources are shown in Figure 2 and Figure 3. In the single dosage treatment, 3-methylsulfinylpropyl isothiocyanate concentrations of 30 nmol·mL^−1^ (χ^2^ = 32.22, df = 4, *p* = 0.001), 20 nmol·mL^−1^ (χ^2^ = 29.20, df = 4, *p* = 0.015), 10 nmol·mL^−1^ (χ^2^ = 25.4, df = 4, *p* = 0.001), and 5 nmol·mL^−1^ (χ^2^ = 17.80, df = 4, *p* = 0.002) were significantly more attractive to adult females than HB (control). However, no significant differences were observed between the control and concentrations of 0.50 nmol·mL^−1^ (χ^2^ = 1.80, df = 4, *p* = 0.205) and 0.01 nmol·mL^−1^ (χ^2^ = 1.40, df = 4, *p* = 0.213) (Figure 2). 

Similarly, in the single treatment of 4-methylsulfinyl-3-butenyl, concentrations of 25 nmol·mL^−1^ (χ^2^ = 21.20, df = 4, *p* = 0.013), 10 nmol·mL^−1^ (χ^2^ = 17.20, df = 4, *p* = 0.007), 5 nmol·mL^−1^(χ^2^ = 20.16, df = 4, *p* = 0.012), and 1 nmol·mL^−1^ (χ^2^ = 17.81, df = 4, *p* = 0.005) had significant effects on adult females as compared to the control, while no significant differences were observed between the control and 0.50 nmol·mL^−1^ (χ^2^ = 5.00, df = 4, *p* = 0.288) and 0.10 nmol·mL^−1^ (χ^2^ = 1.80, df = 4, *p* = 0.205) concentrations (Figure 3).

For the blended treatments, the results showed that the HB blend (χ^2^ = 21.20, df = 4, *p* = 0.001) was more attractive to adult females than the FLB blends (χ^2^ = 4.20, df = 4, *p* = 0.12) or TLB blends (χ^2^ = 9.01, df = 4, *p* = 0.20), while the HB blend (χ^2^ = 21.20, df = 4, *p* = 0.10) and MB blend (χ^2^ = 17.60, df = 4, *p* = 0.15) showed no significant difference (Figure 4).

Sulfur-containing isothiocyanate; 3-methylsulfinylpropyl isothiocyanate, and 4-methylsulfinyl-3-butenyl isothiocyanate volatile compounds from the headspace were significantly correlated with the olfactory response of adult females. Therefore were considered for single and blended bioassays, while the others were not (Table 4).

Adult female responses to the two active chemical compound blends in a Y-olfactometer were compared using the χ2 test. (1) blend HB: 3-methylsulfinylpropyl isothiocyanate 18.71 nmol·mL−1; (2) blend FLB: 3-methylsulfinylpropyl isothiocyanate 8.32 nmol·mL−1 and 4-methylsulfinyl-3-butenyl isothiocyanate 1.36 nmol·mL−1; (3) blend TLB: 3-methylsulfinylpropyl isothiocyanate 22.91 nmol·mL−1 and 4-methylsulfinyl-3-butenyl isothiocyanate 1.89 nmol·mL−1; (4) blend MB: 3-methylsulfinylpropyl isothiocyanate 18.78 nmol·mL−1 and 4-methylsulfinyl-3-butenyl isothiocyanate 1.43 nmol·mL−1. Bars indicate mean ± SE; means followed by different letters indicate significant differences at *p* < 0.05.

## 4. Discussion

Crucifer plants effectively use their secondary metabolites to protect themselves against herbivores. HIPVs are an evolved plants response that stimulates behavioral response in herbivores [25,26,27,28,29,30]. However, host plant recognition by herbivores is dependent on the perception of a specific substance, or combination of substances, at or near the surface of leaves [31]. Therefore, our study results show that volatile constituents (3-methylsulfinylpropyl isothiocyanate and 4-methylsulfinyl-3-butenyl isothiocyanate) may be just as effective as non-volatile constituents (glucosinolates) to oviposit [20,32,33,34,35,36,37]. These results suggested that DBM may be aware of host plants while in flight.

In our study we used *B. vulgaris* which is considered as a trap crop against the DBM [4,5,34,35,38]. Our Y-tube olfactometer bioassay results show that HIPVs emitted from *B. vulgaris* affect the behavior of adult DBM females. We observed that mechanically damaged plants were less attractive to the DBM adult female compared with healthy plants (Table 3). Moreover, it is surprising that plants infested with third instars or healthy plants were more attractive to the DBM adult females than plants infested with first instars (Table 3). Our results also showed that the DBM responded positively to 3-methylsulfinylpropyl isothiocyanate and 4-methylsulfinyl-3-butenyl isothiocyanate, volatile compounds found on the leaf surfaces of crucifer plants [21]. This suggests that the up or down regulation of plant volatiles (isothiocyanate), significantly affect the DBM female attraction to oviposit on *B. vulgaris* plants.

Volatile compounds such as isothiocyanates are also involved in the behavioral response to hydrolysis output of glucosinolates in crucifers [38,39]. However, hydrolysis is expected to occur only when the leaf is infested with insects which can activate isothicyanates [40]. Isothiocyanates in crucifer plants [41,42,43,44,45] have been shown to occur in abundance in the airspace above undamaged plants [46]. Some studies also showed that HIPVs function as host-attraction cues for a range of specialist herbivores [28,29,47]. The host attraction phenomena was observed in by DBM and another crucifer feeders, the cabbage root fly [1,48].These finding suggested that the feeding by early instars decreased attraction of adult DBM females to host plant, which in result may regulates volatiles emission.

Additionally, responses to the synthetic active chemical compounds (Figure 2 and Figure 3) showed that volatile compounds (host-attraction cues) were perceived by olfactory behavior. Previously, it was reported that volatiles from various fertilized crucifer plants play a crucial role in herbivore host preferences [49], although isothiocyanates were not detected in damaged plants (within the threshold limits of the GC-MS system), these findings are echoed in our result that feeding by early instars suppressed emissions.

This study concurs with a previous study on the attractiveness of *B. vulgaris* to ovipositing the DBM [21], where adult females preferentially oviposited on *B. vulgaris* despite the fact that *P. xylostella* larvae do not survive on this dead-end trap crop [50,51,52,53,54,55,56,57,58]. Because, the manipulation of trap crops and their natural enemies has potential applications for pest control in agricultural settings. Thus, the manipulation of *B. vulgaris* could help in controlling DBM.

## 5. Conclusions

Our results provide evidence that the DBM adults can recognize hosts (food odor) in part by using larvae which activates volatile compounds from *B. vulgaris* plants. Our results also showed that host attraction cues (HIPVs) released from *B. vulgaris* damaged by larvae or mechanical means were involved in host attraction. Because when *B. vulgaris* is damaged by larvae it activates secondary metabolites, such as glucosinolates (non-volatile) and isothiocyanates (volatile), which increased the attraction among adult of the DBM females to oviposit. Thus, the results of this study suggested that *B. vulgaris* infested with early instar larvae with insect secretions may alter the amounts and balance of volatiles emitted by the infested plants, thus reducing host attraction. Additional detailed studies on the effect of the DBM-larvae secretions on damaged *B. vulgaris* are needed to explore structure-activity relationship and the sequence of events leading to host attraction.

## Figures and Tables

**Figure 1 insects-11-00725-f001:**
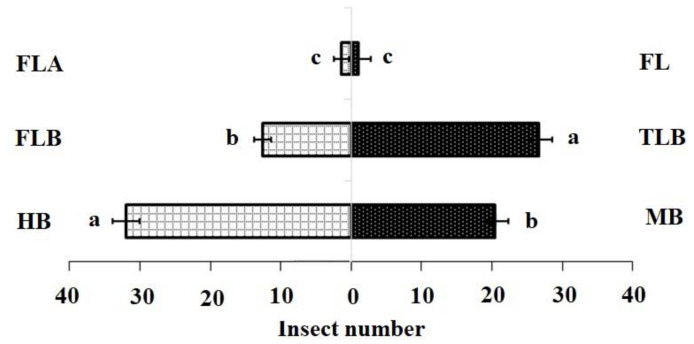
Olfactory responses of *Plutella xylostella* adults to natural volatiles. Bars represent the mean (±SE) response of adult female (compared using the χ^2^ test) exposed to natural volatiles from *Barbarea vulgaris* plants. Different letters indicate differences between two treatments at *p* < 0.05. HB, healthy *B. vulgaris*; FLB, first instar larvae feeding on *B. vulgaris*; TLB, third instar larvae feeding on *B. vulgaris*; MB, mechanically damaged *B. vulgaris*; FLA, first instar larvae feeding on artificial diet; FL, first instar larvae only (FL).

**Figure 2 insects-11-00725-f002:**
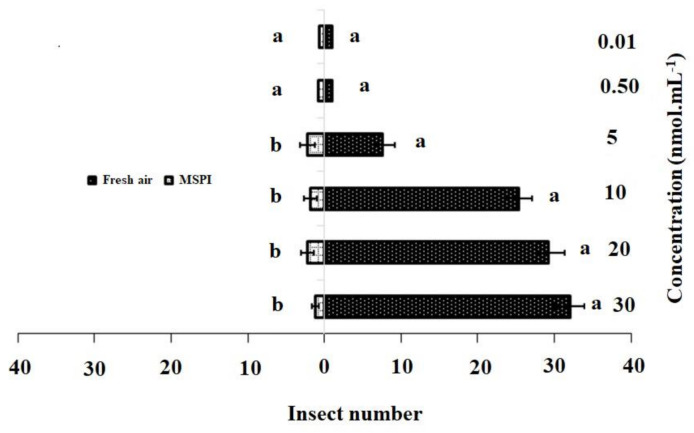
Responses of *P. xylostella* adult females to synthetic 3-methylsulfinylpropyl isothiocyanate (MSPI) compared with control. Bars represent the mean (± SE) response of adult female exposed to synthetic active volatile from *Barbarea vulgaris* plants. Different letters indicate differences between two treatments at *p* < 0.05.

**Figure 3 insects-11-00725-f003:**
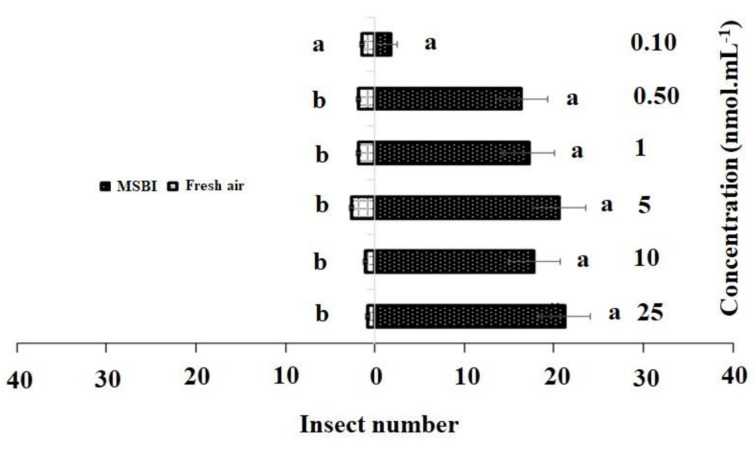
Responses of *P. xylostella* adult females to synthetic 4-methylsulfinyl-3-butenyl isothiocyanate (MSBI) compared with controls. Bars represent the mean (± SE) response of adult female exposed to synthetic active volatile from Barbarea vulgaris plants. Different letters indicate differences between two treatments at *p* < 0.05.

**Figure 4 insects-11-00725-f004:**
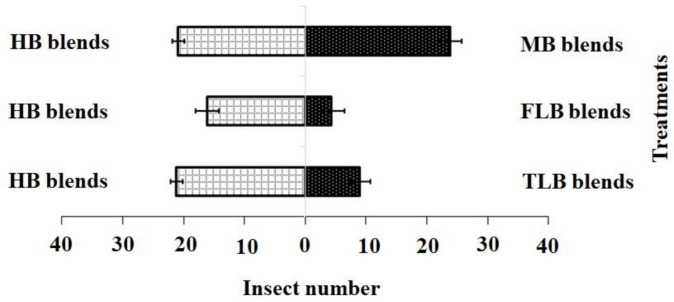
Responses of *P. xylostella* adult females to synthetic blends. HB, healthy *B. vulgaris*; FLB, first instar larvae feeding on *B. vulgaris*; TLB, third instar larvae feeding on *B. vulgaris*; MB, mechanically damaged *B. vulgaris.* Bars represent the mean (± SE) response of adult female exposed to synthetic active volatile from *Barbarea vulgaris* plants. Different letters indicate differences between two treatments at *p* < 0.05.

**Table 1 insects-11-00725-t001:** Mean quantities (ng) ± SE of isothiocyanate from the headspace of multiple damaged *Barbarea vulgaris* plants.

Treatment *	Major Active Compounds (ng)	
	3-Methylsulfinylpropyl Isothiocyanate	4-Methylsulfinyl-3-Butenyl Isothiocyanate
HB	2.09 ± 0.16 ^b^	20.13 ± 2.4 ^a^
FLB	0.12 ± 0.03 ^c^	1.56 ± 0.04 ^c^
TLB	3.51 ± 0.28 ^a^	3.55 ± 0.55 ^c^
MB	2.18 ± 0.31 ^b^	12.01 ± 2.47 ^b^

* HB, healthy *B. vulgaris*; FLB, first instar larvae feeding on *B. vulgaris*; TLB, third instar larvae feeding on *B. vulgaris*; MB, mechanically damaged *B. vulgaris.* Means followed by different letter within a column represent significant differences at *p* < 0.05.

**Table 2 insects-11-00725-t002:** The peak area of the main volatile compounds emitted from multi-treated *Barbarea vulgaris* plants.

Main Compounds	Peak Area × 10^−6^ pA·s (μL/mL)			
	HB *	FLB	TLB	MB
hexahydrofarnesyl acetone	4.18	5.38	10.21	3.53
*n*-heptanal	4.19	4.74	0.17	3.59
*α*-pinene	2.34	5.07	2.92	3.16
hexadecanoic acid	12.20	9.97	15.62	11.28
while phytol	2.59	2.9	0.14	2.75
(*Z*)-3-hexenyl acetate	7.67	7.78	6.36	8.3
sabinene	2.8	2.99	2.52	3.15
3-methylsulfinylpropyl isothiocyanate	26.64	1.63	7.36	24.9
4-methylsulfinyl-3-butenyl isothiocyanate	1.68	0.65	0.45	1.88

* HB, healthy *B. vulgaris*; FLB, first instar larvae feeding on *B. vulgaris*; TLB, third instar larvae feeding on *B. vulgaris*; MB, mechanically damaged *B. vulgaris.*

**Table 3 insects-11-00725-t003:** Olfactory response of the DBM adult females to volatile blends emitted from multi-treated *Barbarea vulgaris* plants.

Treatment *	Mean Moths (%) ± SD **	95% Confidence Interval
HB	32.00 ± 1.949 ^a^	26.58–37.41
FLB	14.80 ± 1.855 ^d^	9.65–19.95
TLB	20.40 ± 1.778 ^c^	15.46–25.33
MB	26.60 ± 1.931 ^b^	21.28–31.91

* HB, healthy *B. vulgaris*; FLB, first instar larvae feeding on *B. vulgaris*; TLB, third instar larvae feeding on *B. vulgaris*; MB, mechanically damaged *B. vulgaris*. ** Means followed by the same letter within the column are not significantly different at *p* < 0.05.

**Table 4 insects-11-00725-t004:** Correlation between olfactory response of adult females to the multi-treated *Barbarea vulgaris* plant volatiles and peak area of the main volatile compounds emitted from multi-treated *B. vulgaris.*

Main Compounds	Correlation Coefficients between Compounds and Activity of Volatile Blends	
	Pearson Correlation	Sig. (2-Tailed)
hexahydrofarnesyl acetone	−0.455	0.55
*n*-heptanal	0.122	0.88
*α*-pinene	−0.861	0.14
hexadecanoic acid	−0.665	0.34
while phytol	−0.181	0.82
(*Z*)-3-hexenyl acetate	−0.272	0.73
sabinene	0.052	0.95
3-methylsulfinylpropyl isothiocyanate	0.962 *	0.04
4-methylsulfinyl-3-butenyl isothiocyanate	0.825 *	0.18

* Correlation coefficients with significant differences.

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
