# Peer review of "Diamondback Moth Larvae Trigger Host Plant Volatiles that Lure Its Adult Females for Oviposition"

_insects, 2020, doi:10.3390/insects11110725_

Round 1
Reviewer 1 Report
The suggestions are comments for the authors are provided in the attached manuscript pdf.

Author Response
Response to Reviewer 1 Comments (Minor revision)
Manuscript number: insects-972580
Manuscript title: Diamondback Moth Larvae Trigger Host Plant Volatile that Lure its Adult Females for Oviposition
The authors would like to thank the Editor and Reviewer 1 for his/her quick and deep review of our manuscript with very constructive comments and suggestions. The general and specific questions or concerns and recommendations are highly appreciated, and corrections have been made accordingly in the revised version of the manuscript to improve its quality. The point-by-point responses to the reviewer 1 comments and suggestions are listed as follows:
Comments and Suggestions for Authors:
The suggestions are comments for the authors are provided in the attached manuscript pdf.
Response: Many thanks to Reviewer 1 for his / her detailed and constructive review. All the comments in attached PDF are critically analyzed and addressed, and suggestions incorporated to improve the quality of the manuscript.
Reviewer 2 Report
Generally, this MS is well revised.
---
The originality of this MS is not clear. There are lot of previous study in this field. For example, Shiojiri and Takabayashi (2003: Ecol. Entomol. 28: 573–578) reported the oviposition preference of DBM when offered a choice between an uninfested cabbage plant and a plant infested with DBM larvae. And Choh et al . (2008: Ecological Entomology, 33: 565–568) also described the detail. Please check well the previous studies and make clear the originality of this MS. I think the abstract, discussion and conclusions should be greatly improved.
Response: We thank the reviewer 2 for the straightforward comments but we disagree with the reviewer 2 on the originality (novelty) of our study. Our study originality (novelty) is totally different than a) Shiojiri and Takabayashi (2003: Ecol. Entomol. 28: 573–578) and b) Choh et al . (2008: Ecological Entomology, 33: 565–568), as referred by Reviewer 2. Their objective of study were as follows;
- a) To clarify reasons for the oviposition preferences of xylostella, the performance of host larvae on cabbage plants infested with conspecific larvae, flight responses of C. plutellae towards two plants with differing levels of host‐larvae‐induced damage, and oviposition numbers of the wasp on infested plants were studied.
- b) The present study examined the oviposition response of xylostella to plants infested by parasitized P. xylostella larvae. The study also examined the flight response of C. vestalis to plants infested by parasitised and non‐parasitised larvae, and the performance of P. xylostella on plants infested with either parasitised or non‐parasitised larvae as possible factors to explain the oviposition preference of P. xylostella.
But our objective of study was:
-We quantified different volatiles (natural and simulated) released by brassica plants in response to the DBM larval feeding. We investigated olfactory responses of adult the DBM to natural and simulated volatiles released by infested plants. We also studied how volatile effects changed in response to different larval feeding treatments.
-and we have tested these behavioural responses on a DBM trap crop instead of common host plants such as cabbage.
Regarding this, I mentioned the two papers as example. Now the objectives and originality were well described.
L35: “±1.855” and “±1.778” are SE or SD? In addition, please delete “;”.
Response: These are SD and we have already mentioned this in the Table 5. Deleted as suggested. Please see table 5 and line 36 in revised version
Even though authors described in Table, it’s hard to understand for the potential readers. Please delete the data. Or please explain the meaning of the number at abstract too.
Author Response
Response to Reviewer 2 Comments (Minor revision)
Manuscript number: insects-972580
Manuscript title: Diamondback Moth Larvae Trigger Host Plant Volatile that Lure its Adult Females for Oviposition
The authors would like to thank the Editor and Reviewer 2 for his/her quick and deep review of our manuscript with very constructive comments and suggestions. The general and specific questions or concerns and recommendations are highly appreciated, and corrections have been made accordingly in the revised version of the manuscript to improve its quality. The point-by-point responses to the reviewer 2 comments and suggestions are listed as follows:
Comments and Suggestions for Authors:
- Generally, this MS is well revised.
The originality of this MS is not clear. There are lot of previous study in this field. For example, Shiojiri and Takabayashi (2003: Ecol. Entomol. 28: 573–578) reported the oviposition preference of DBM when offered a choice between an uninfested cabbage plant and a plant infested with DBM larvae. And Choh et al . (2008: Ecological Entomology, 33: 565–568) also described the detail. Please check well the previous studies and make clear the originality of this MS. I think the abstract, discussion and conclusions should be greatly improved.
Response: We thank the reviewer 2 for the straightforward comments but we disagree with the reviewer 2 on the originality (novelty) of our study. Our study originality (novelty) is totally different than a) Shiojiri and Takabayashi (2003: Ecol. Entomol. 28: 573–578) and b) Choh et al . (2008: Ecological Entomology, 33: 565–568), as referred by Reviewer 2. Their objective of study were as follows;
- a) To clarify reasons for the oviposition preferences of xylostella, the performance of host larvae on cabbage plants infested with conspecific larvae, flight responses of C. plutellae towards two plants with differing levels of host‐larvae‐induced damage, and oviposition numbers of the wasp on infested plants were studied.
- b) The present study examined the oviposition response of xylostella to plants infested by parasitized P. xylostella larvae. The study also examined the flight response of C. vestalis to plants infested by parasitised and non‐parasitised larvae, and the performance of P. xylostella on plants infested with either parasitised or non‐parasitised larvae as possible factors to explain the oviposition preference of P. xylostella.
But our objective of study was:
-We quantified different volatiles (natural and simulated) released by brassica plants in response to the DBM larval feeding. We investigated olfactory responses of adult the DBM to natural and simulated volatiles released by infested plants. We also studied how volatile effects changed in response to different larval feeding treatments.
-and we have tested these behavioural responses on a DBM trap crop instead of common host plants such as cabbage.
- Regarding this, I mentioned the two papers as example. Now the objectives and originality were well described.
Response: Many thanks to Reviewer 2 for his / her concerns and constructive review.
L35: “±1.855” and “±1.778” are SE or SD? In addition, please delete “;”.
Response: These are SD and we have already mentioned this in the Table 5. Deleted as suggested. Please see table 5 and line 36 in revised version
- Even though authors described in Table, it’s hard to understand for the potential readers. Please delete the data. Or please explain the meaning of the number at abstract too.
Response: We have explained the meaning of the numbers at abstract. Please see L34-35 in the revised version.
This manuscript is a resubmission of an earlier submission. The following is a list of the peer review reports and author responses from that submission.
Round 1
Reviewer 1 Report
The title of the study needs minor revision to ‘Diamondback Moth Larvae Trigger Host Plant Volatiles that Lure its Adult Females for Oviposition’
The study’s goals, objectives and hypothesis need to be inserted clearly.
Simple summary, abstract, introduction, materials & methods and results looks fine, however, discussion and conclusions need improvements. I need authors to focus on what has been determined in the results based on the HIPVs to clearly explain their effects on the behavior of the plant feeding stages (larvae tested) and egg laying females. Also, the induced mechanical damage to the plant leaves and damage due to the larval feeding need to be discussed further with respect to female DBM’s behavior. Finally, the conclusion section needs additional work. It needs to be explained in detail for example to include other compounds measured and their effects on the behavior of the moths.
The following are the other suggestions/comments provided in the manuscript’s text:
Line #s Comments/suggestions
21 before Diamondback insert ‘the’
change cruciferous to crucifer
22 delete importance and insert ‘attention’
23 delete of which competing before host plants
after plants insert which are
24 insert ‘to the pest’ after attractive
25 delete also
26 delete studied and insert investigated
27 delete different
28 change indicate to indicated
29-31 insert ‘the’ before the diamondback
32 change volatile to volatiles
34 and onward use two digits instead of three throughout the text and tables
36 change isothiocyanates to isothiocyanate
38 delete comma and insert ‘while’
39 change demonstrates to demonstrated that
40 & onward Insert ‘the’ before the DBM and diamondback where indicated
43 change Crucifers to Crucifer crops
delete biological control
44 insert oviposition after organic compounds
47 delete a before risk and insert ‘s’ in the parenthesis after the risk
Delete herbivorous and ‘s’ in the insects and insert herbivores
56 change cruciferous to crucifer and keep it uniform throughout the text
60 delete which, and insert which after plants
Insert ‘to the pest’ after attractive
61 Rephrase this sentence “Plant volatiles plays a significant role in the acceptance of potential host plants, and have been widely studied in Lepidoptera”
63 delete generalist and insert ‘polyphagous’
64 change lepidoptera to Lepidoptera
65 delete herbivore induced plant volatiles and remove parenthesis from HIPVs
69 delete also
76 & onward remove spaces between number and °C
80, 83, 107 delete % after 60
81 change bassica to brassica
87 delete treatments after response
89 insert Control after HB
93-94 delete Plants of B. vulgaris were 60-70 days old when assessed.
94 delete ‘and’ after replicates and insert ‘for’
delete was considered a replicate
97 delete herbivore induced plant volatiles and remove parenthesis in HIPVs
157, 185 change Barbarea to B.
164 change behavioral to behavioural
181 Insert how correlation analysis was carried out
195 insert ‘the’ before first
229, 239 Figure 2 & 3. Enhance the resolution of the graph legend
269-272 use brassica or crucifer throughout the text
271 delete in result and insert ‘can’ if this make any sense
277 delete specialist and insert ‘feeder’
290 delete and their natural enemies
219 after B. vulgaris insert and ‘its natural enemies’
292 after acceptance insert ‘of’
294-296 delete this text “Authors should discuss the results and how they can be interpreted in perspective of previous studies and of the working hypotheses. The findings and their implications should be discussed in the broadest context possible. Future research directions may also be highlighted”
317-482 Reference sections need proper formatting according to Journal’s guidelines. For example., species name should be Italicized and Journal (sources) names should be modified to title levels.

Author Response
Response to Reviewer 1 Comments
Manuscript number: insects- 921050
Manuscript title: Diamondback Moth Larvae Trigger Host Plant Volatile that Lure its Adult Females
The authors would like to thank the Editor and Reviewer 1 for his/her quick and deep review of our manuscript with very constructive comments and suggestions. The general and specific questions or concerns and recommendations are highly appreciated, and corrections have been made accordingly in the revised version of the manuscript to improve its quality. The point-by-point responses to the reviewer 1 comments and suggestions are listed as follows:
Comments and Suggestions for Authors:
The title of the study needs minor revision to ‘Diamondback Moth Larvae Trigger Host Plant Volatiles that Lure its Adult Females for Oviposition’
Response: We have made minor changes in the title as suggested. Please see title in revised version.
The study’s goals, objectives and hypothesis need to be inserted clearly.
Response: We have captured goals, objectives and hypothesis in the revised version of manuscript. Please see lines 68-72 in the revised version
Simple summary, abstract, introduction, materials & methods and results looks fine, however, discussion and conclusions need improvements. I need authors to focus on what has been determined in the results based on the HIPVs to clearly explain their effects on the behavior of the plant feeding stages (larvae tested) and egg laying females. Also, the induced mechanical damage to the plant leaves and damage due to the larval feeding need to be discussed further with respect to female DBM’s behavior. Finally, the conclusion section needs additional work. It needs to be explained in detail for example to include other compounds measured and their effects on the behavior of the moths.
Response: We have revise discussion and conclusion section to improve the quality of manuscript. Please see revised version.
The following are the other suggestions/comments provided in the manuscript’s text:
Line #s Comments/suggestions
21 before Diamondback insert ‘the’ change cruciferous to crucifer
Response: Inserted “The” before Diamondback and changed cruciferous to crucifer. Please see line 21in revised version.
22 delete importance and insert ‘attention’
Response: Deleted and inserted ‘attention’. Please see line 22 in the revised version.
23 delete of which competing before host plants
Response: Deleted ‘of which competing’ before host plants. Please see line 23-24 in the revised version.
after plants insert which are
Response: Inserted “which” after plants. Please see line 24 in the revised version.
24 insert ‘to the pest’ after attractive
Response: Inserted “to the pest” after attractive. Please see line 24 in the revised version.
25 delete also
Response: Deleted “also”. Please see line 25 in the revised version.
26 delete studied and insert investigated
Response: Deleted “studied” and inserted “investigated”. Please see line 27 in the revised version.
27 delete different
Response: Deleted “different”. Please see line 28 in the revised version.
28 change indicate to indicated
Response: We have changed “indicate” to “indicated”. Please see line 28 in the revised version.
29-31 insert ‘the’ before the diamondback
Response: Inserted “The” as suggested. Please see line 31 in the revised version.
32 change volatile to volatiles
Response: Changed “volatile” to “volatiles”. Please see line 32 in the revised version.
34 and onward use two digits instead of three throughout the text and tables.
Response: Made changes in text and tables as suggested. Please see revised version.
36 change isothiocyanates to isothiocyanate
Response: Changed isothiocyanates to isothiocyanate. Please see line 36 in the revised version.
38 delete comma and insert ‘while’
Response: deleted comma and inserted ‘while’. Please see line 38 in the revised version.
39 change demonstrates to demonstrated that
Response: Changed demonstrates to “demonstrated that”. Please see line 40 in the revised version.
40 & onward Insert ‘the’ before the DBM and diamondback where indicated
Response: Done as suggested. Please see revised version.
43 change Crucifers to Crucifer crops
Response: Changed Crucifers to “Crucifer crops”. Please see line 45 in the revised version.
delete biological control
Response: Deleted as suggested. Please see line 45 in the revised version.
44 insert oviposition after organic compounds
Response: Inserted oviposition keyword after organic compounds. Please see line 46 in the revised version.
47 delete a before risk and insert ‘s’ in the parenthesis after the risk
Response: Done as suggested. Please see line 49 in the revised version.
Delete herbivorous and ‘s’ in the insects and insert herbivores
Response: Deleted herbivorous and ‘s’ in the insects and inserted “herbivores”. Please see line 49 in the revised version.
56 change cruciferous to crucifer and keep it uniform throughout the text
Response: Done as suggested. Please see revised version.
60 delete which, and insert which after plants
Response: Deleted which, and inserted which after plants. Please see line 62 in the revised version.
Insert ‘to the pest’ after attractive
Response: Inserted ‘to the pest’ after attractive. . Please see line 62 in the revised version.
61 Rephrase this sentence “Plant volatiles plays a significant role in the acceptance of potential host plants, and have been widely studied in Lepidoptera”
Response: Rephrased as suggested. Please see line 63-64 in the revised version.
63 delete generalist and insert ‘polyphagous’
Response: Deleted generalist and insert ‘polyphagous’. Please see line 66 in the revised version.
64 change lepidoptera to Lepidoptera
Response: Done as suggested. Please see line 68 in the revised version.
65 delete herbivore induced plant volatiles and remove parenthesis from HIPVs
Response: Done as suggested. Please see line 69 in the revised version.
69 delete also
Response: Deleted as suggested. Please see line 74 in the revised version.
76 & onward remove spaces between number and °C
Response: Done as suggested. Please see revised version.
80, 83, 107 delete % after 60
Response: Deleted % after 60. Please see line 86 in the revised version.
81 change bassica to brassica
Response: Sorry for unseen text. We have changed bassica to brassica. Please see line 87 in the revised version.
87 delete treatments after response
Response: Deleted treatments as suggested. Please see line 95 in the revised version.
89 insert Control after HB
Response: We have added control as suggested. Please see line 95 in the revised version.
93-94 delete Plants of B. vulgaris were 60-70 days old when assessed.
Response: We have deleted this line to improve the quality of our manuscript.
94 delete ‘and’ after replicates and insert ‘for’
Response: Deleted ‘and’ after replicates and inserted ‘for’ as suggested. Please see line 100 in the revised version.
delete was considered a replicate
Response: Deleted as suggested. Please see line 100 in the revised version.
97 delete herbivore induced plant volatiles and remove parenthesis in HIPVs
Response: Deleted as suggested. Please see line 103 in the revised version.
157, 185 change Barbarea to B.
Response: Abbreviated as suggested. Please see lines 164 and 192 in the revised version.
164 change behavioral to behavioural
Response: Changed behavioral to behavioural as suggested. Please see line 171 in the revised version.
181 Insert how correlation analysis was carried out
Response: We ahave added correlation analyses information in “Statistical analysis” section. Please see lines 191-192 in the revised version.
195 insert ‘the’ before first
Response: Inserted ‘the’ before first. Please see line 200 in the revised version.
229, 239 Figure 2 & 3. Enhance the resolution of the graph legend
Response: We have enhanced the resolution of the graph legends. Please see Figure 2 and 3 in revised version.
269-272 use brassica or crucifer throughout the text
Response: We prefer crucifer and revised as suggested. Please see line 281 in the revised version.
271 delete in result and insert ‘can’ if this make any sense
Response: We have inserted ‘can’ to improve sentence. Please see line 283 in the revised version.
277 delete specialist and insert ‘feeder’
Response: Deleted as suggested and inserted feeder. Please see line 289.in the revised version.
302 delete and their natural enemies
Response: Deleted as suggested.
219 after B. vulgaris insert and ‘its natural enemies’
Response: Revised as suggested. Please see line 302 in the revised version.
292 after acceptance insert ‘of’
Response: Inserted of as suggested. Please see line 304 in the revised version.
294-296 delete this text “Authors should discuss the results and how they can be interpreted in perspective of previous studies and of the working hypotheses. The findings and their implications should be discussed in the broadest context possible. Future research directions may also be highlighted”
Response: Sorry for unseen text. Deleted as suggested.
317-482 Reference sections need proper formatting according to Journal’s guidelines. For example, species name should be Italicized and Journal (sources) names should be modified to title levels.
Response: We have revised the reference list. Please see revised version of manuscript.
Reviewer 2 Report
The originality of this MS is not clear. There are lot of previous study in this field. For example, Shiojiri and Takabayashi (2003: Ecol. Entomol. 28: 573–578) reported the oviposition preference of DBM when offered a choice between an uninfested cabbage plant and a plant infested with DBM larvae. And Choh et al . (2008: Ecological Entomology, 33: 565–568) also described the detail. Please check well the previous studies and make clear the originality of this MS. I think the abstract, discussion and conclusions should be greatly improved.
L35: “±1.855” and “±1.778” are SE or SD? In addition, please delete “;”.
L81: When the DBM larvae were collected?
L97: How the mated and unmated females were identified?
L294-296: I think no need.
Table 1 and 2: These tables are not directly related to major subject of this paper. Please consider moving the tables to supplementary material.
Author Response
Response to Reviewer 2 Comments
Manuscript number: insects- 921050
Manuscript title: Diamondback Moth Larvae Trigger Host Plant Volatile that Lure its Adult Females
The authors would like to thank the Editor and Reviewer 2 for his/her quick and deep review of our manuscript with very constructive comments and suggestions. The general and specific questions or concerns and recommendations are highly appreciated, and corrections have been made accordingly in the revised version of the manuscript to improve its quality. The point-by-point responses to the reviewer 2 comments and suggestions are listed as follows:
Comments and Suggestions for Authors
The originality of this MS is not clear. There are lot of previous study in this field. For example, Shiojiri and Takabayashi (2003: Ecol. Entomol. 28: 573–578) reported the oviposition preference of DBM when offered a choice between an uninfested cabbage plant and a plant infested with DBM larvae. And Choh et al . (2008: Ecological Entomology, 33: 565–568) also described the detail. Please check well the previous studies and make clear the originality of this MS. I think the abstract, discussion and conclusions should be greatly improved.
Response: We thank the reviewer 2 for the straightforward comments but we disagree with the reviewer 2 on the originality (novelty) of our study. Our study originality (novelty) is totally different than a) Shiojiri and Takabayashi (2003: Ecol. Entomol. 28: 573–578) and b) Choh et al . (2008: Ecological Entomology, 33: 565–568), as referred by Reviewer 2. Their objective of study were as follows;
- a) To clarify reasons for the oviposition preferences of xylostella, the performance of host larvae on cabbage plants infested with conspecific larvae, flight responses of C. plutellae towards two plants with differing levels of host‐larvae‐induced damage, and oviposition numbers of the wasp on infested plants were studied.
- b) The present study examined the oviposition response of xylostella to plants infested by parasitized P. xylostella larvae. The study also examined the flight response of C. vestalis to plants infested by parasitised and non‐parasitised larvae, and the performance of P. xylostella on plants infested with either parasitised or non‐parasitised larvae as possible factors to explain the oviposition preference of P. xylostella.
But our objective of study was:
-We quantified different volatiles (natural and simulated) released by brassica plants in response to the DBM larval feeding. We investigated olfactory responses of adult the DBM to natural and simulated volatiles released by infested plants. We also studied how volatile effects changed in response to different larval feeding treatments.
-and we have tested these behavioural responses on a DBM trap crop instead of common host plants such as cabbage.
L35: “±1.855” and “±1.778” are SE or SD? In addition, please delete “;”.
Response: These are SD and we have already mentioned this in the Table 5. Deleted as suggested. Please see table 5 and line 36 in revised version
L81: When the DBM larvae were collected?
Response: DBM larvae were collected from the cabbage and cauliflower in 2014 and 2016, and they have been reared for more than 50 generations since then.
L97: How the mated and unmated females were identified?
Response: Mating success was inferred by means of the existence of DBM male made scale patches on the abdomen of DBM females.
L294-296: I think no need.
Response: Sorry for unseen text. Deleted as suggested. Please see line 307-309 in the revised version.
Table 1 and 2: These tables are not directly related to major subject of this paper. Please consider moving the tables to supplementary material.
Response: We have moved Tables 1 and 2 to supplementary material (Tables S1 and S2) as suggested. Please see revised version.
Reviewer 3 Report
The manuscript is about the attractive effect of B. vulgaris volatiles for DBM adults in laboratory tests. Results are interesting regarding the fact the adults are more attracted to volatiles released by larvae infested plants and synthetic compounds in comparison to volatiles released by healthy plants. In addition, older larvae seem to cause more damage and induce a higher release of volatiles than younger ones therefore those plants were more attractive to adults.
The manuscript has its credits but there are many questions unanswered in the methodology and results that should be proper address before publication.
First of all, keywords (and discussion) include biological control, but this by any means was studied in this research.
Some references cited throughout the text are not related to the sentence itself, for instance, L.67 (refs: 11, 14, 15) are not related to natural enemies' attraction to HIPVs. Also, L.61 (refs 13, 18) are not related to Lepidoptera behavior of plant selection.
Methods
Section 2.3 – very confusing. What are the dependent and independent variables? How many replicates per treatment? Why did not test the response of adults to plant treatments against the clean air (control)? What is the real difference between FLA and FL, if all larvae were reared on an artificial diet? Why did not compare TLA x TL?
Younger larvae cause significant less amount of leaf damage than older larvae. Why did not use an insect equivalent (% damage) in order to standardize the amount of damage and induction of HIPVs on plants? The same amount of larvae in different instar does not account for that.
2.5 - Synthetic compounds tested in different concentrations, but what is the equivalence of those concentrations to the real amount released by the infested plants? Tables show a very different concentration between synthetic compounds released by the cotton pad and that released by the plants. Why did not contrast the synthetic compounds against the clean air (control)?
- 108 – Authors mentioned using a black box to conduct tests and fluorescent light. DBM is a crepuscular insect and light might have some impact on adult behavior. Why not conducted the tests in a dark room with red light? What time did the bioassays were run?
Statistics
Data on Table 5 is compared among treatments within the columns, but those are independent tests, therefore cannot be compared.
Results
L.232-234 – Compounds were not repellent, but attractive.
L.243- P = 0.2 (HB x TLB), this is not statistically significant, therefore was not different.
Fig. 4. Bars are not followed by letters
Line 261-262 – Results are presented in a table, not in a figure. There are no bars to refer to.
Table 6. seg. 0.175 – P > 0.05, there is no significant difference.
Discussion
Should try to explain the results found and what are the implications of those results for DBM ecology and management in Brassica crops.
- 268 – the compound exo-2-acetyl-5-iso… was not detected not tested in this study.
- 286 If you did not control for the amount of damage caused by younger and older larvae on leaves you cannot affirm younger larvae suppress the release of volatiles in B. vulgaris plants.
- 290. Pure speculation since you did not measure anything related to natural enemies’ responses in this study.
- 294-296 – Text does not belong to the manuscript content. It seems to be copied from the instructions for authors.
Host acceptance, which comprises feeding and/or oviposition, was not accessed in this study. The authors only measured host attraction in lab experiments. Therefore, should refrain from this kind of affirmation. One could only imply that additional tests might investigate those aspects in order to testify that this plant is really more accepted that Brassica crops.
Author Response
Response to Reviewer 3 Comments
Manuscript number: insects- 921050
Manuscript title: Diamondback Moth Larvae Trigger Host Plant Volatile that Lure its Adult Females
The authors would like to thank the Editor and Reviewer 3 for his/her quick and deep review of our manuscript with very constructive comments and suggestions. The general and specific questions or concerns and recommendations are highly appreciated, and corrections have been made accordingly in the revised version of the manuscript to improve its quality. The point-by-point responses to the reviewer 3 comments and suggestions are listed as follows:
Comments and Suggestions for Authors
The manuscript is about the attractive effect of B. vulgaris volatiles for DBM adults in laboratory tests. Results are interesting regarding the fact the adults are more attracted to volatiles released by larvae infested plants and synthetic compounds in comparison to volatiles released by healthy plants. In addition, older larvae seem to cause more damage and induce a higher release of volatiles than younger ones therefore those plants were more attractive to adults.
The manuscript has its credits but there are many questions unanswered in the methodology and results that should be proper address before publication.
Response: We would like to thank Reviewer 3 for nice remark and pointing out the gaps to be attended in the underlined sections of the manuscript. Efforts have been made to revisit these sections and corrections have been done accordingly as requested in the revised version of manuscript.
First of all, keywords (and discussion) include biological control, but this by any means was studied in this research.
Response: We used ‘biological control’, because DBM oviposit extensively on Barbarea vulgaris, despite its larvae do not survive on this crop and its proposed as a trap crop in previous studies (Badenes-Perez et al., 2004,. 2005; Lu et al., 2004). But since our results do not drive directly to that, we decided to delete this keyword to avoid any confusion.
Some references cited throughout the text are not related to the sentence itself, for instance, L.67 (refs: 11, 14, 15) are not related to natural enemies' attraction to HIPVs.
Response: We have removed ‘and attracting predators’ to correlated other references and improve the quality of our manuscript. Please see line 71 in the revised version.
Also, L.61 (refs 13, 18) are not related to Lepidoptera behavior of plant selection.
Response: Sentence now reformulated to improve the quality of our manuscript. Please see L.64-65 in the revised version.
Methods
Section 2.3 – very confusing. What are the dependent and independent variables?
Response: We have revised this section to improve the quality of manuscript. Please see L.103-118 in the revised version.
How many replicates per treatment?
Response: In total, there were five replicates in this experiment. We have added this information the revised version of manuscript to avoid any confusion. Please see L.138 in the revised version.
Why did not test the response of adults to plant treatments against the clean air (control)?
Response: Healthy (HB) was used as control (clean air) in this experiment. We added this information to avoid confusion. Please see L.105 in the revised version.
What is the real difference between FLA and FL, if all larvae were reared on an artificial diet?
Response: We did not rear all larvae on artificial diet.
Why did not compare TLA x TL?
Response: Because most of first instar larvae failed to survive, when fed on B. vulgaris.
Younger larvae cause significant less amount of leaf damage than older larvae. Why did not use an insect equivalent (% damage) in order to standardize the amount of damage and induction of HIPVs on plants? The same amount of larvae in different instar does not account for that.
Response: We fully agree with the Reviewer 3 about using insect equivalent (% damage) in order to standardize the amount of damage and induction of HIPVs on plants. However, this prediction objective underlined by Reviewer 3, is one of the multiple objectives of a Big Project which is tackling the management of DBM under different geographical zones and conditions. Other studies (objectives) are warranted to capture this aspect of the project.
2.5 - Synthetic compounds tested in different concentrations, but what is the equivalence of those concentrations to the real amount released by the infested plants?
Response: We have predicted concentrations of synthetic compounds which are the equivalence of those concentrations to the real amount released by the infested plants. Please see Table S2 in the revised version of manuscript.
Tables show a very different concentration between synthetic compounds released by the cotton pad and that released by the plants. Why did not contrast the synthetic compounds against the clean air (control)?
Response: We have predicted concentrations of synthetic compounds which are the equivalence of those concentrations to the real amount released by the infested plants and Healthy (HB) was used as control (clean air) in this experiment.
108 – Authors mentioned using a black box to conduct tests and fluorescent light. DBM is a crepuscular insect and light might have some impact on adult behavior. Why not conducted the tests in a dark room with red light? What time did the bioassays were run?
Response: DBM Females were released at 17.00 h, and we used dark room as females preferred to oviposit in the dark (Harcourt, 1957); “Harcourt, D.G. (1957) Biology of the diamondback moth, Plutella maculipennis (Curt.) (Lepidoptera: Plutellidae), in eastern Ontario. II. Life‐history, behaviour, and host relationships. Canadian Entomologist, 89, 554–564.”
Statistics
Data on Table 5 is compared among treatments within the columns, but those are independent tests, therefore cannot be compared.
Response: We have performed Pearson Correlation analysis to compare among treatments.
Results
L.232-234 – Compounds were not repellent, but attractive.
Response: We have rephrased this sentence. Please see L.262 in the revised version.
L.243- P = 0.2 (HB x TLB), this is not statistically significant, therefore was not different.
Response: HB (P = 0.001) is more attractive to adult DBM females than TLB (P = 0.2). Please see Figure 5 as well.
Fig. 4. Bars are not followed by letters
Response: We have revised Figure 4 and improved the quality of figure. Please see revised version of manuscript.
Line 261-262 – Results are presented in a table, not in a figure. There are no bars to refer to.
Response: Please see Figure 4 for your attention.
Table 6. seg. 0.175 – P > 0.05, there is no significant difference.
Response: In Table 6. seg. 0.175 is a correlation coefficients.
Discussion
Should try to explain the results found and what are the implications of those results for DBM ecology and management in Brassica crops.
Response: We have revised this section and clear conclusion points are now stated as regards to the findings of the study.
- 268 – the compound exo-2-acetyl-5-iso… was not detected not tested in this study.
Response: We have tested 3-methylsulfinylpropyl isothiocyanate and 4-methylsulfinyl-3-butenyl isothiocyanate which are analog of exo-2-acetyl-5-isothiocyanatonorbornane
- 286 If you did not control for the amount of damage caused by younger and older larvae on leaves you cannot affirm younger larvae suppress the release of volatiles in B. vulgaris plants.
Response: We have compared FLB and TLB with mechanically damaged B. vulgaris plants, thus we think there no need to control for the amount of damage caused by younger and older larvae on leaves.
- 290. Pure speculation since you did not measure anything related to natural enemies’ responses in this study.
Response: We have rephrased this sentence to avoid any confusion. Please see L.320 in the revised version.
- 294-296 – Text does not belong to the manuscript content. It seems to be copied from the instructions for authors.
Response: Sorry for unseen text. Deleted as suggested. Please see line 307-309 in the revised version.
Host acceptance, which comprises feeding and/or oviposition, was not accessed in this study. The authors only measured host attraction in lab experiments. Therefore, should refrain from this kind of affirmation. One could only imply that additional tests might investigate those aspects in order to testify that this plant is really more accepted that Brassica crops.
Response: We have changed ‘host acceptance’ to ‘host attraction’ to avoid such confusion.